# Bio-Mitigation of Carbon Dioxide Using *Desmodesmus* sp. in the Custom-Designed Pilot-Scale Loop Photobioreactor

**Abhishek Anand** [1], **Kaustubh Tripathi** [1], **Amit Kumar** [1], **Suresh Gupta** [1], **Smita Raghuvanshi** [1,*] and **Sanjay Kumar Verma** [2]

1 Department of Chemical Engineering, Birla Institute of Technology and Science (BITS), Pilani 333031, Rajasthan, India; p20170014@pilani.bits-pilani.ac.in (A.A.); f2014538@pilani.bits-pilani.ac.in (K.T.); h20170029@pilani.bits-pilani.ac.in (A.K.); sureshg@pilani.bits-pilani.ac.in (S.G.)

2 Department of Biological Sciences, Birla Institute of Technology and Science (BITS), Pilani 333031, Rajasthan, India; skverma@pilani.bits-pilani.ac.in

* Correspondence: smita@pilani.bits-pilani.ac.in; Tel.: +91-1596-255638

**Abstract:** Today's society is faced with many upfront challenges such as the energy crisis, water pollution, air pollution, and global warming. The greenhouse gases (GHGs) responsible for global warming include carbon dioxide ($CO_2$), methane ($CH_4$), nitrous oxide ($NO_x$), water vapor ($H_2O$), and fluorinated gases. A fraction of the increased emissions of $CO_2$ in the atmosphere is due to agricultural and municipal solid waste (MSW) management systems. There is a need for a sustainable solution which can degrade the pollutants and provide a technology-based solution. Hence, the present work deals with the custom design of a loop photobioreactor with 34 L of total volume used to handle different inlet $CO_2$ concentrations (0.03%, 5%, and 10% (*v/v*)). The obtained values of biomass productivity and $CO_2$ fixation rate include $0.185 \pm 0.004$ g $L^{-1}$ $d^{-1}$ and $0.333 \pm 0.004$ g $L^{-1}$ $d^{-1}$, respectively, at 10% (*v/v*) $CO_2$ concentration and $0.084 \pm 0.003$ g $L^{-1}$ $d^{-1}$ and $0.155 \pm 0.003$ g $L^{-1}$ $d^{-1}$, respectively, at 5% (*v/v*) $CO_2$ concentration. The biochemical compositions, such as carbohydrate, proteins, and lipid content, were estimated in the algal biomass produced from $CO_2$ mitigation studies. The maximum carbohydrate, proteins, and lipid content were obtained as $20.7 \pm 2.4\%$, $32.2 \pm 2.5\%$, and $42 \pm 1.0\%$, respectively, at 10% (*v/v*) $CO_2$ concentration. Chlorophyll (Chl) a and b were determined in algal biomass as an algal physiological response. The results obtained in the present study are compared with the previous studies reported in the literature, which indicated the feasibility of the scale-up of the process for the source reduction of $CO_2$ generated from waste management systems without significant change in productivity. The present work emphasizes the cross-disciplinary approach for the development of bio-mitigation of $CO_2$ in the loop photobioreactor.

**Keywords:** *Desmodesmus* sp.; loop photobioreactor; biomass productivity; $CO_2$ fixation rate; lipid

## 1. Introduction

Global warming caused by rising carbon dioxide ($CO_2$) emissions is currently a worldwide concern. Since industrialization, global greenhouse gas (GHG) emissions have increased due to human activities [1]. The primary source of $CO_2$ emissions includes anthropogenic waste, fossil fuel combustion, transportation, municipal waste, and agriculture waste [2,3]. Most GHG emissions generated from agricultural waste occur through the various waste management stages and agricultural inputs, mainly from water, fertilizers, pesticides from the soil, residue management, and irrigation [4]. Another sector, municipal solid waste management, significantly contributes to GHG emissions, mainly $CO_2$, methane ($CH_4$), and nitrous oxide ($N_2O$). From collection to treatment and disposal, the waste management process must be optimized to reduce greenhouse gas emissions [5]. One of the previous reports suggested that the anthropogenic emission of $CO_2$ from municipal waste and the agriculture sector is responsible for global $CO_2$ emissions up to 3.2% and

18.4%, respectively [6]. According to one of the recent reports, GHG emissions from the agricultural sector, including livestock such as cows, agricultural soils, and rice production, accounted for 10% of the total GHG emissions [7].

Thus, GHG emissions from municipal solid waste and agriculture waste treatment methods have raised concerns about climate change [8,9]. $CO_2$ is one of the most significant GHG emissions. The estimated emission of $CO_2$ in 2014 was 6870 MMT (million metric tons), which contributes to around 81% of the total GHG emissions in the world [10]. Currently, $CO_2$ concentration is above 400 ppm (parts per million) according to the data obtained from the NOAA Earth System Research Laboratory, Global Monitoring Division [11]. The projected concentration of $CO_2$ will rise to the value of 600 ppm, resulting in the rise of sea level from 0.4 to 1 m. It can also lead to ocean acidification in the twenty-first century [12,13]. The Intergovernmental Panel on Climate Change (IPCC) stated that if the appropriate action is not taken to prevent the continual increase of GHG emissions, the earth's temperature will increase by 1.4–5.8 °C during the 21st century [14,15]. Hence, carbon capture sequestration (CCS) and carbon capture utilization (CCU) strategies are utilized to cut down the $CO_2$ emissions from sources [16,17]. The present research is emphasized in the domain of CCU [18].

The physical and chemical methods to mitigate $CO_2$ emissions include absorption, cryogenic separation, ionic liquids, and $CO_2$ storage [19,20]. However, these approaches entail higher energy consumption, construction, and operating costs [21,22]. In recent times, biological methods of $CO_2$ mitigation gained the attention of researchers due to the production of biomass energy during $CO_2$ fixation by photosynthetic processes [23,24]. Photosynthetic microorganisms such as microalgae have an efficiency of 10–50 times higher than terrestrial plants, with a CO2 fixation rate between 0.73 and 2.22 g $L^{-1}$ $day^{-1}$ [25,26]. The microalgae-based mitigation process has several advantages, such as a higher growth rate than terrestrial plants [27] and completes the recycling of $CO_2$. $CO_2$ is converted into biomass via photosynthesis activity by utilizing nitrogen and phosphorous as a nutrient source and solar energy as an energy source, which can be further transformed into fuels using existing technologies. Later, fuels can be utilized to produce power and result in $CO_2$ formation [28,29].

Conventionally, algae can be cultivated in an open culture system (raceway ponds) or a closed system (photobioreactors). A study carried out on 1 L glass made in a closed photobioreactor for the bio-mitigation of $CO_2$ by *Scendesmus obliquious* reported the $CO_2$ consumption rate values as 390.2 mg $L^{-1}d^{-1}$ [30]. Another study discussed the $CO_2$ fixation by *Scenedesmus* sp. in a closed photobioreactor having dimensions of 33 cm length and 4.5 cm inner diameter [31]. The study demonstrated an integrated system for $CO_2$ fixation from flue gas, wastewater remediation, and biomass production. Similarly, few studies have reported the $CO_2$ mitigation study on raceway pond (open pond) systems. Raceway ponds are utilized for $CO_2$ mitigation for the large-scale cultivation of algae species such as *Chlorella*, Dunaliella, and *S. platensis*, [30,32]. The raceway ponds are the best examples of open pond systems due to better nutrient mixing and biomass sedimentation. The disadvantages of raceway ponds are that, compared to the closed photobioreactor, raceway ponds show lower productivity because of the carbon limitation [32].

Closed photobioreactors mostly give higher biomass productivities and also prevent outside contamination [33]. Given the benefits of closed systems over open ponds, various photobioreactors (from laboratory to industrial scale) have been developed. Even though many photobioreactors have been studied, only a small number of these reactors can efficiently use solar energy for mass algal production. The majority of outdoor photobioreactors, such as flat-plate, horizontal, and inclined tubular photobioreactors, have exposed lightning surfaces. Bubble-column, airlift, and stirred-tank photobioreactors offer high scalability, but their application in outdoor cultures is limited due to their low illumination surface areas [33,34]. While many photobioreactors appear simple to run at the laboratory scale, only a few photobioreactors have been successfully scaled up at the pilot scale. The difficulties in maintaining optimal light, temperature, mixing, and mass transfer in photo-

bioreactors make these scale-up techniques extremely difficult. The absence of effective photobioreactors is one of the primary blockades to mass algae production.

Overcoming these limitations, loop bioreactors are efficient reactors that provide uniform and good mixing without mechanical agitation and ease of operation. These are mainly constructed of transparent materials such as glass, plexiglass, polyvinylchloride (PVC), acrylic PVC, or polyethylene [35,36]. Loop reactors are cylindrical vessels that perform the mixing of multiphase fluids without the impeller action. The advantages of the loop reactor are better mixing without impellers and an adequate illumination surface, which allow these reactors to overcome the limitations of flat-plate, horizontal, and inclined tubular photobioreactors. Another significant advantage is that the cost of impellers is not incurred in these loop reactors, leading to energy savings [37,38]. The literature studies indicated that the application of loop bioreactors for $CO_2$ mitigation using microalgae is limited. Most of the studies are confined to bench-scale reactors. Hence, there is an enormous scope to utilize the pilot-scale closed-loop photobioreactor for $CO_2$ fixation using microalgae.

The present study focuses on the bio-mitigation of $CO_2$ in the atmosphere by *Desmodesmus* species in the closed-loop photobioreactor (custom design) of a scale of around 34 L, which is almost a pilot-scale reactor. Bio-mitigation experiments using *Desmodesmus* species were carried out at three different $CO_2$ concentrations, including 0.03% (atmospheric $CO_2$), 5%, and 10% (*v/v*) in the loop photobioreactor. The work includes estimating growth kinetic parameters such as cell concentration, specific growth rate, biomass productivity, and $CO_2$ fixation rate. The biochemical properties, such as chlorophyll content, lipid content, carbohydrate content, protein, and cells, were determined for the obtained biomass. Thus, an optimized process was developed to effectively utilize $CO_2$ generated from waste and in actual day-to-day conditions. This is an economical and alternative source of carbon for the simultaneous production of biomass feedstock rich in lipids and carbohydrates in a "waste to wealth" chain and waste management for sustainable future development.

## 2. Materials and Methods

### 2.1. Media Preparation, Microalgae Strain, Culture Conditions, and Inoculum Preparation

BG-11 Medium was used as cultivation media for the growth of algae, which contains 0.04 g $L^{-1}$ dipotassium hydrogen phosphate ($K_2HPO_4$), 0.006 g $L^{-1}$ of citric acid ($C_6H_8O_7$), 0.006 g $L^{-1}$ of ferric ammonium citrate ($C_6H_{5+4y}FE_xNyO_7$), 0.001 g $L^{-1}$ EDTA, 1.5 g $L^{-1}$ of sodium nitrate ($NaNO_3$), 0.075 g $L^{-1}$ of magnesium sulfate ($MgSO_4.7H_2O$), 0.036 g $L^{-1}$ of calcium chloride ($CaCl_2$), 0.002 g $L^{-1}$ of sodium carbonate ($Na_2CO_3$), and 1 mL $L^{-1}$ of trace metal mix. Trace metal mix comprised boric acid, zinc sulfate, copper sulfate, sodium molybdate, cobalt nitrate, and manganese chloride. The solid media was used for plating and was prepared by adding 1.5 g $L^{-1}$ (1.5%) agar to aqueous media. The conditioned media was autoclaved at 121 °C and 15 psi for 15 min and was used for further studies.

Microalgae strain *Desmodesmus* sp. MCC34 [KF731760.1] was used in the present work [39]. It was collected from the Environment and Microbiology lab of BITS Pilani, Pilani Campus, Rajasthan. The strain was isolated from the local water bodies of Pilani, Rajasthan, as reported by Nagappan and Verma [35].

The inoculum was grown at a constant temperature of 26 ± 1 °C in the laboratory and the light intensity of 67 μmol photon $m^{-2}$ $s^{-1}$ for ten days. The purity of the grown culture was checked using repeated streaking of the culture on BG-11 plates. The cultures with an optical density close to unity were used as inoculum for conducting the experiments in a custom-designed loop photobioreactor at larger volumes. The optical density of culture was measured at a wavelength of 650 nm (OD) at [$OD_{650\,nm}$] (Evolution 201, Thermo Scientific, is Waltham, MA USA) to determine the cell concentration using a UV-Vis spectrophotometer. The calibration curve was prepared between the dry weight of biomass versus optical density to measure the cell concentration.

### 2.2. Experimental Setup

The schematic of loop photobioreactor with detailed custom design of experimental setup is given in Figure 1.

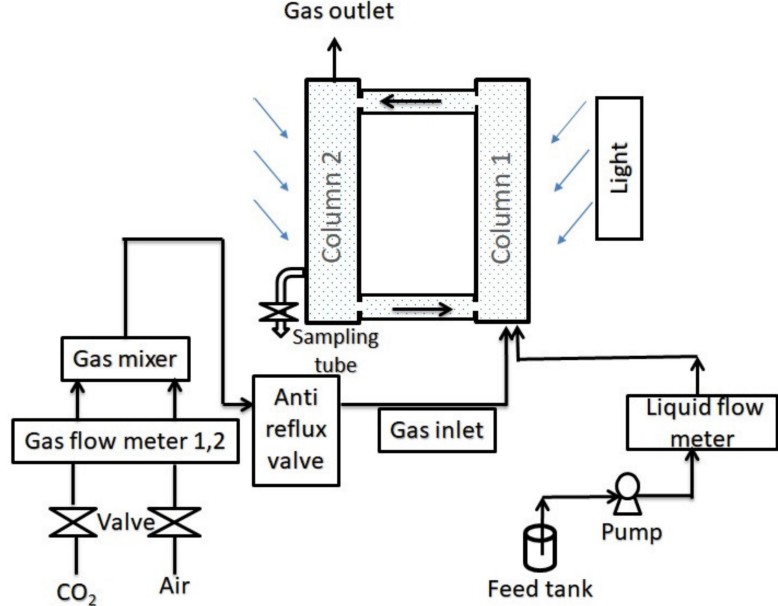

**Figure 1.** Schematic diagram of custom-designed loop photobioreactor.

The loop photobioreactor was constructed with two units and dimensions of 2.03 m, including 0.105 m diameter and 0.12 m outer diameter. The loop photobioreactor with a total volume of 34 L and a working volume of 26 L was designed for the overall process, and a photograph of the actual setup is given in Figure 2. The sunlight was used as the energy source during the process. In this study, 1.25 L of enriched culture was used as inoculum volume, and it had an optical density (OD) of 0.82. The experiments were performed in the semicontinuous mode.

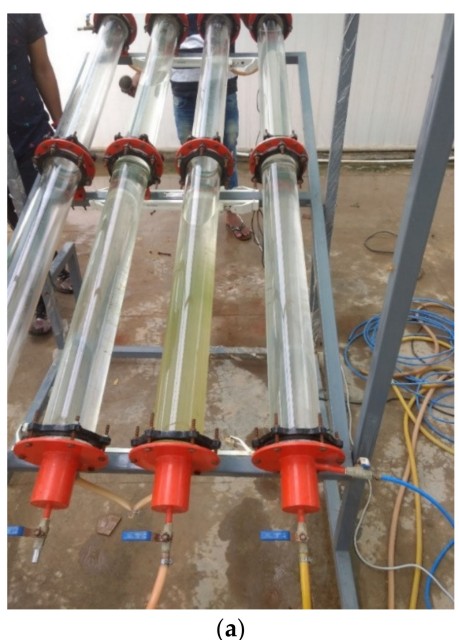
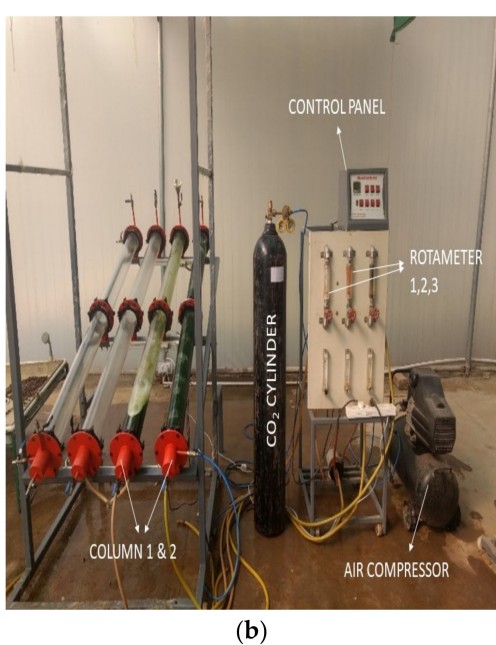

(**a**)                                                                  (**b**)

**Figure 2.** An overview of the loop photobioreactor: (**a**) at day zero after inoculation; (**b**) at day 12—last day of incubation period used for the $CO_2$ mitigation study.

### 2.3. Experimental Procedure

A gas mixture comprising of 10% (*v/v*) $CO_2$ (g) and 90% of compressed moisture-free air was utilized as a source of $CO_2$ (g) in the photobioreactor. The gas mixture was supplied on a 12 h aeration cycle through the gas inlet port of the photobioreactor equipped with the sparger. The gas mixture was fed during the light period, and its supply was stopped during the dark period. The continuous study was performed in a loop photobioreactor for 12 days, and the temperature was maintained at ambient conditions of 30–35 °C. The initial pH was maintained between 7 and 9 for the optimal growth of *Desmodesmus* sp. that increased the solubility of $CO_2$ in the aqueous phase. The flow rate of gas into the reactor was maintained at 32 vvm (4 L min$^{-1}$).

Once the microalgae reached the stationary phase, culture was separated by filtering with muslin cloth and was rinsed with distilled water. The algal biomass was freeze-dried and preserved at −20 °C for carrying out further studies. The parameters such as pH, dry weight biomass, and CFU were measured after every 24 h duration. The optical density ($OD_{650nm}$) was measured twice a day (after completion of the light cycle and a dark process). The control run was performed using ambient air (0.03% $CO_2$) while keeping other conditions the same. The change in color of columns 1 and 2 in picture (b) concerning picture (a) shows *Desmodesmus* sp. after 12 days of the incubation period.

### 2.4. Measurement of Biomass Growth Rate

The dry weight biomass and optical density were measured to evaluate the biomass yield of *Desmodesmus* sp. Fifty milliliters of aliquot culture was collected, and dry weight biomass (g L$^{-1}$) was measured using the standard filtration process [40,41]. The filtrate obtained was utilized for further studies. The contamination was checked by plating the supernatant and colony-forming unit (CFU) method. Aliquots were withdrawn from the loop photobioreactor every 24 h, and pH was measured using a digital pH meter (Eco Testr pH 2, Eutech Instruments).

### 2.5. Determination of Growth Kinetic Parameters

The biomass productivity (*P*) was calculated by the given Equation (1):

$$P = \frac{X_t - X_0}{t_t - t_0} \tag{1}$$

where $X_t$ is the cell concentration (g L$^{-1}$) at the end of the cultivation cycle ($t_t$), and $X_0$ is the initial cell concentration (g L$^{-1}$) at $t_0$ (day). The specific growth rate $\mu_{max}$ (day$^{-1}$) was calculated using Equation (2) [42,43].

$$\mu_{max} = \frac{\ln N_2 - \ln N_1}{t_2 - t_1} \tag{2}$$

where $N_1$ and $N_2$ are the concentrations of the cells at the beginning ($t_1$) and the end ($t_2$) of the exponential growth phase, respectively [43,44]. $C_1O_{0.48} H_{1.83}N_{0.11}P_{0.01}$ was used as the microalgal biomass molecular formula stated in previous studies [45]. As per the reported studies, it is assumed that 1 g of produced algal biomass ($C_1O_{0.48} H_{1.83}N_{0.11}P_{0.01}$) is equivalent to capture 1.88 g of $CO_2$, and hence, the $CO_2$ fixation rate (g L$^{-1}$ d$^{-1}$) was determined from Equation (3) [31,46].

$$\text{Fixation rate of } CO_2 = 1.88 \times P_{\text{overall}} \tag{3}$$

where $P_{\text{overall}}$ is the overall biomass productivity. $CO_2$ utilization efficiency was obtained from Equation (4).

$$CO_2 \text{ utilization efficiency} = \frac{\text{fixatation rate of } CO_2}{CO_{2_{in}}} \times 100 \tag{4}$$

### 2.6. Determination of Chlorophyll Content

Fifty milliliters of algal culture sample was collected in the falcon tube and was centrifuged for 10 min at 4000 rpm. The supernatant was discarded, and the pellet was stored at −20 °C until further use. During the extraction step, the pellet was re-suspended in 90% methanol. It was further assisted by sonication for cell lysis under dark conditions in the ice bath (to prevent the degradation of chlorophyll from light). The control parameters followed during the sonication were of 1 min timer and a 60% duty cycle. The thermal shock was given by the snap freezing method in liquid nitrogen, and the whole process was repeated for ten cycles to maximize the extraction yield. This step was followed by centrifugation at 4000 rpm for 10 min, and the pellet was dried and stored. One milligram of dried algal biomass was taken in the falcon tube, and a mixture of 90% methanol and 10% Millipore water was added to maintain the volume of 10 mL. The tube was kept in the water bath for 20 min, and then it was stored at 4 °C for the incubation period of 24 h. The absorbance of the obtained supernatant was measured at 652 and 665 nm in a spectrophotometer. Methanol was used as a blank solution in a UV-Vis spectrophotometer. The concentration of Chl a and Chl b was determined according to the following Equations (5) and (6) [46].

$$\text{Chl a }\left(\text{mgL}^{-1}\right) = (16.72 \times \text{absorbance}_{665nm}) - (9.16 \times \text{absorbance}_{652nm}) \tag{5}$$

$$\text{Chl b }\left(\text{mgL}^{-1}\right) = (34.09 \times \text{absorbance}_{652nm}) - (15.28 \times \text{absorbance}_{665nm}) \tag{6}$$

### 2.7. Biochemical Compositional Analysis

Biomass collected after every sampling point (as explained in Section 2.4) has been utilized for biochemical compositional analysis.

#### 2.7.1. Analysis of Total Carbohydrate (CHO) Content

A 5 mL sample was taken and centrifuged at 5 °C for 10 min at 4000 rpm, and the obtained supernatant was discarded. The pellets were washed with deionized water and stored at −20 °C for further studies. Then, 0.5 mL of 2.5 M $H_2SO_4$ was added in the pellet to carry out primary hydrolysis (polysaccharides to monosaccharides) [47,48]. The samples were placed for incubation in a boiling water bath for two hours. The columns were cooled at room temperature, and hydrolysate was diluted with deionized water to make it to the volume of 5 mL. The particular step was followed by centrifugation at 4000 rpm, and the supernatant was collected. The phenol-sulfuric method was applied to determine the total content of carbohydrates in biomass [49]. The calibration plot was drawn at different glucose concentrations (0–0.1 mg mL$^{-1}$). Two–milliliter aliquots of diluted supernatant along with standard solution were mixed with 1 mL of 5% aqueous phenol in a 15 mL falcon tube. Then, 5 mL of concentrated sulfuric acid was immediately added in all tubes and then vortexed for 10 s. All falcon tubes were kept at room temperature for 10 min, and then these were placed in the water bath at 30 °C to develop a yellow-golden color. The value of absorbance was measured at 490 nm in a UV-Vis spectrophotometer [50].

#### 2.7.2. Analysis of Total Protein Content

The Folin–Lowry method was used for the total protein determination using white pellets obtained after pigment extraction [51]. The pellet was pretreated with 1% SDS/0.1 M NaOH in 500 μL. The re-suspended pellet mixed with reagent A (500 μL of 1:1:1:1 ratio of CTC (0.1% $CuSO_4 \cdot 5H_2O$ + 0.2% NaK tartrate +10% $Na_2CO_3$), 10% SDS, 0.8 M NaOH and $dH_2O$) and the tubes were kept at room temperature for 10 min. After adding reagent B (250 μL of a solution of one volume of Folin–Ciocalteu reagent and five volumes of $dH_2O$) to the samples, tubes were instantly vortexed and allowed to stand at room temperature for 30 min. The OD was measured at 750 nm for 0.5 mL of 1% SDS/0.1 M NaOH. The standard curve was prepared for the determination of the total amount of protein by

dissolving different concentrations of bovine serum albumin (BSA) in 1% SDS/0.1 M NaOH (0–1.0 mg mL$^{-1}$) as reported by Varshney et al. (2016) [52].

### 2.7.3. Analysis of Total Lipid Content

The total lipid content of the biomass was quantified gravimetrically using the Bligh and Dyer method with slight modifications [53]. The pellets were separated from the 50 mL culture after centrifugation at 4000 rpm for 10 min at 4 °C and were stored at −20 °C for further studies. The pellet was suspended in 2.4 mL deionized water followed by 3 mL chloroform and 6 mL methanol. It was followed by sonication by placing the mixture in the ultrasonic bath for 20 min. Further, 3 mL of each deionized water and chloroform were added to maintain the final ratio of 2:2:1.8. The final mixture was centrifuged for 10 min at 2000 rpm. The organic bottom layer of chloroform was carefully extracted after centrifugation and was transferred into a pre-weighted vial and preserved overnight for solvent evaporation in the fume hood. The vial was reweighted until dry to determine the overall lipid quantity, and these steps were carried out at room temperature as per the procedure reported by Varshney et al. (2018) [50].

### 3. Results and Discussion

The semicontinuous studies analyzed the growth performance of *Desmodesmus* sp. for 12 days, for the three different $CO_2$ concentrations, 0.03%, 5%, and 10% (*v/v*), in a loop photobioreactor. The obtained results from these studies were analyzed and summarized in the following sections.

### 3.1. Effect of $CO_2$ Concentration on Biomass Growth Rate and Optical Density Values

*Desmodesmus* sp. growth performance in the presence of different $CO_2$ (0.03%, 5%, and 10% *v/v*) concentrations was examined in a loop photobioreactor. During all experiments, microalgae showed a short lag phase of 1 to 3 days, which indicated the suitability of gaseous $CO_2$ mitigation by *Desmodesmus* sp. as a carbon source, as shown in Figure 3. The trend shows the increased growth in the exponential phase for ten days at different concentrations of $CO_2$ due to the presence of the appropriate amount of nutrient for cell growth in the reactor.

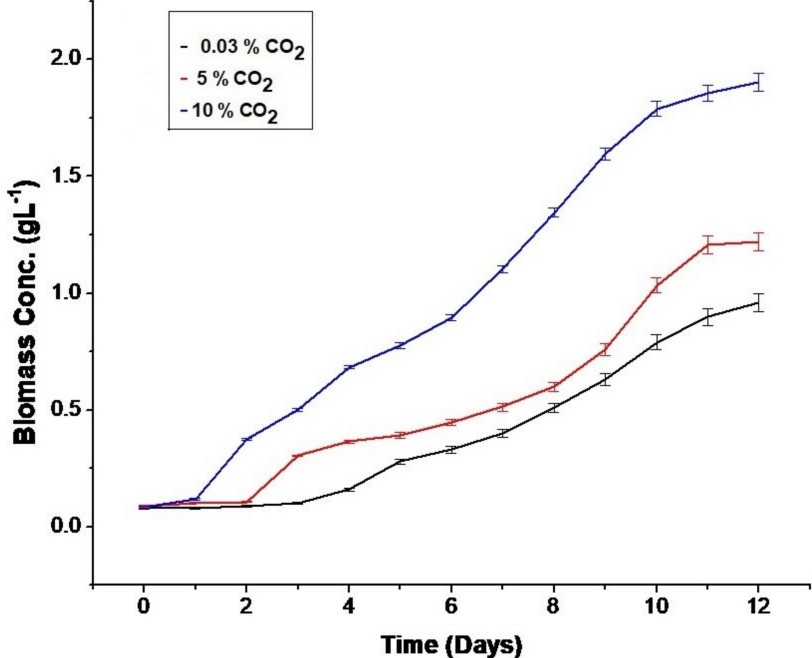

**Figure 3.** Effect of time on cell concentration (g L$^{-1}$) at three different $CO_2$ concentrations (0.03%, 5%, and 10% *v/v*).

After 12 days of incubation, culture supplemented with 10% (*v/v*) $CO_2$ showed $1.903 \pm 0.038$ g $L^{-1}$ of cell concentration on a dry cell weight (DCW) basis, which is higher as compared to the culture grown at 0.03% of $CO_2$ *v/v* ($0.96 \pm 0.039$ g $L^{-1}$) and at 5% $CO_2$ *v/v* ($1.219 \pm 0.040$ g $L^{-1}$). The cell concentration in cultures supplied with 5% $CO_2$ and 10% $CO_2$ was higher than the cultures with ambient air conditions, suggesting that $CO_2$ as a carbon source facilitated microalgae growth [54,55]. A similar trend was reported for $CO_2$ mitigation in earlier reported studies [56,57].

The results are also plotted to understand the light and dark cycle (L/D cycle) on biomass growth rate in optical density ($OD_{650nm}$) at three different $CO_2$ concentrations and are given in Figure 4. The values of optical density were measured twice a day [58].

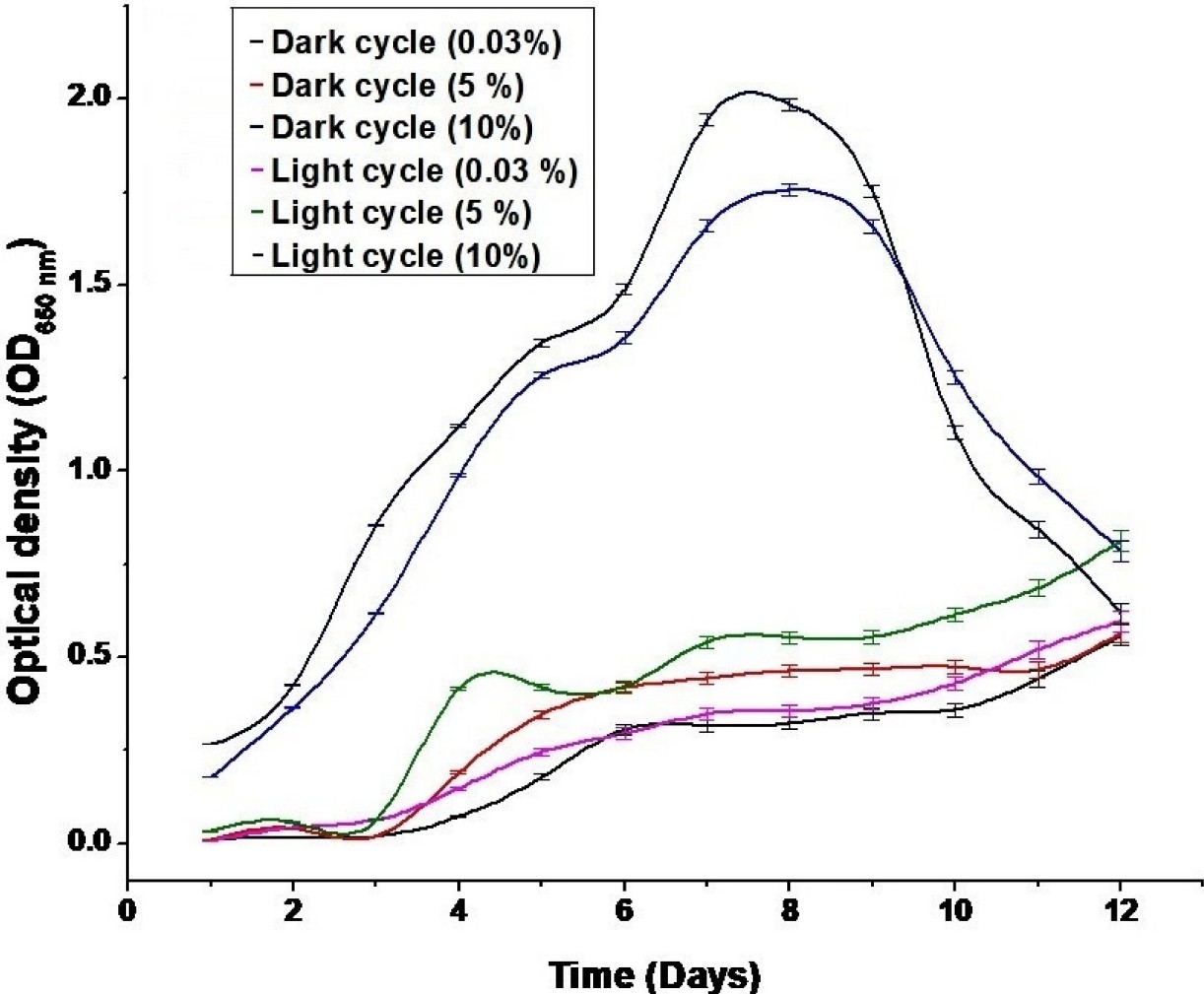

**Figure 4.** Effect of light and dark cycle on optical density (OD) with respect to time for three different $CO_2$ concentrations.

The maximum OD values were obtained during the light cycle as $0.60 \pm 0.017$, $0.81 \pm 0.016$, and $1.99 \pm 0.010$ and during the dark cycle as $0.56 \pm 0.016$, $0.57 \pm 0.017$, and $1.76 \pm 0.009$ at 0.03% $CO_2$, 5% $CO_2$, and 10% $CO_2$ concentrations, respectively. The increased absorbance values during the light cycle compared to the dark cycle confirmed that the photosynthesis process is enhanced during the day and microalgal growth is better during the light period. The increase in the biomass concentration values during the day cycle enhances the understanding that the increased biomass concentration is due to the increased cell growth, and hence is greatly dependent on the sunlight intensity [59].

It has been reported by the researchers that the photosynthetic efficiency of microalgae under intermittent illumination is known to be higher than under continuous illumination, provided that the parameters of the L/D cycle are tuned correctly [60,61]. The reasoning could be that photosynthesis is a cyclic process, where a slower thermochemical process follows almost instantaneous photochemical reactions.

### 3.2. Effects of CO₂ Concentration on Growth Kinetic Parameters

### 3.2.1. Specific Growth Rate

The specific growth rate ($\mu$) of algal culture was measured using Equation (2) as given in Section 2.5. The maximum value of $\mu_m$ was obtained as $0.15 \pm 0.004$ d$^{-1}$ when algal cells were grown with 10% inlet $CO_2$ concentration. The specific growth rate was observed to be $0.07 \pm 0.002$ d$^{-1}$ and $0.13 \pm 0.003$ d$^{-1}$ for 0.03% and 5% inlet $CO_2$ concentration, respectively (Figure 5). The marginal difference in the value of $\mu$ was observed to change $CO_2$ concentration from 5% to 10%. These results are as per the reported results in the earlier studies [57,62].

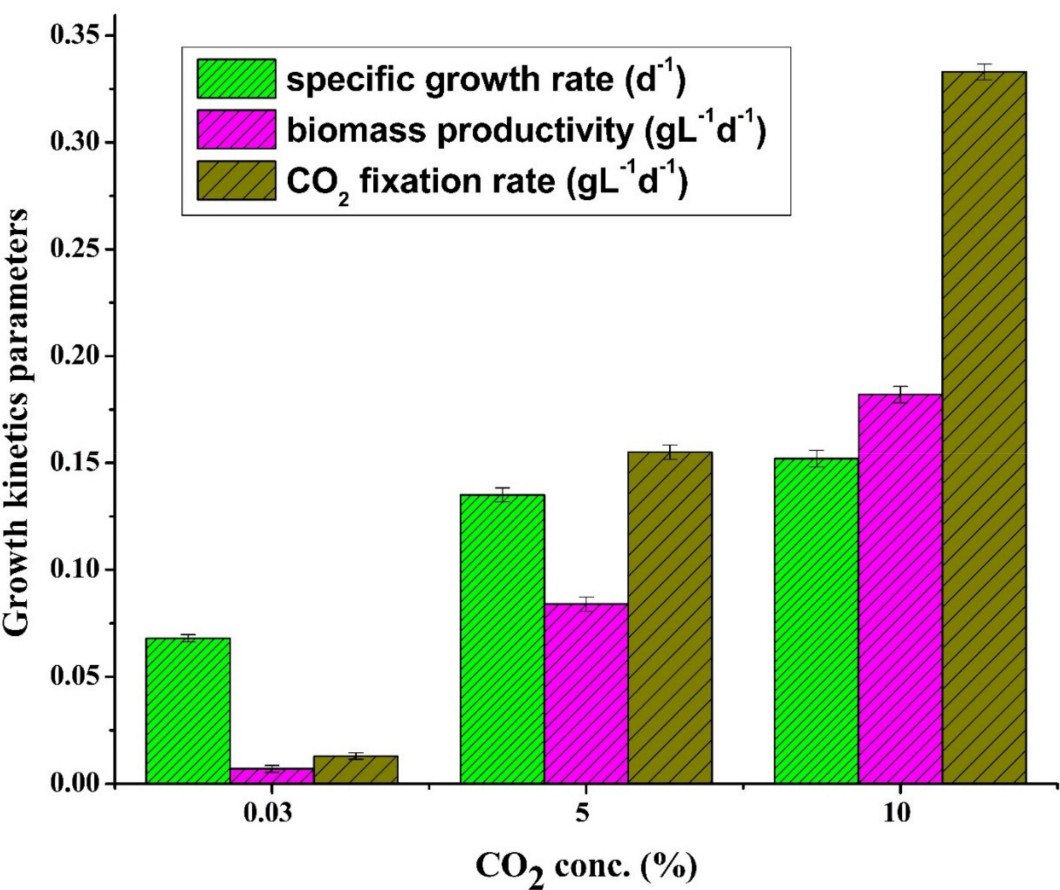

**Figure 5.** Effect of three different $CO_2$ concentrations (0.03%, 5%, and 10% *v/v*) on the different growth kinetic parameters (specific growth rate (d$^{-1}$), biomass productivity (g L$^{-1}$ d$^{-1}$), $CO_2$ fixation rate (g L$^{-1}$ d$^{-1}$)) of microalgae.

### 3.2.2. Biomass Productivity

The concentration of $CO_2$ significantly influences the productivity of biomass. The biomass productivity was estimated for all three inlet concentrations and is shown in Figure 5. It was observed that with the increase in $CO_2$ concentration from 0.03% to 10%, the biomass productivity value was increased from $0.018 \pm 0.002$ g L$^{-1}$ d$^{-1}$ to $0.185 \pm 0.004$ g L$^{-1}$ d$^{-1}$. These findings are consistent with values obtained by studies given by [45,62].

### 3.2.3. $CO_2$ Fixation Rate

The $CO_2$ fixation rate was calculated using Equation (3). It was observed that the higher rate of biofixation of $CO_2$ ($0.33 \pm 0.004$ g $L^{-1}$ $d^{-1}$) was achieved when microalgae were cultured at 10% inlet concentration of $CO_2$ (Figure 5). The $CO_2$ fixation rates were obtained as $0.01 \pm 0.001$ g $L^{-1}$ $d^{-1}$ and $0.15 \pm 0.003$ g $L^{-1}$ $d^{-1}$ for 0.03% and 5% of $CO_2$ concentration, respectively. These results are supported by the work carried out by different researchers [57,63].

### 3.3. Effect of $CO_2$ Concentration on Biochemical Composition of Desmodesmus sp.

The content of lipids, total carbohydrates, proteins, and chlorophyll was estimated as macromolecular composition in the form of percentages of the total dry biomass (DCW) at three different $CO_2$ concentrations (0.03%, 5%, and 10% *v/v*). The concentration of $CO_2$ has a significant impact on the carbohydrate (CHO) content of microalgae [52]. CHO content of microalgae was observed as $14.6 \pm 1.5$%, $17.2 \pm 2.0$%, and $20.7 \pm 2.4$% of DCW for 0.03% $CO_2$, 5% $CO_2$, and 10% $CO_2$ (*v/v*) concentration, respectively (Table 1). The different stages of growth and varying concentrations of $CO_2$ have a greater impact on the total content of carbohydrates in the harvested algal biomass. The carbohydrate content in the algal biomass was significantly increased with an increase in $CO_2$ concentration. The higher content of carbohydrates opens the possibility for further utilization of algal biomass as a substrate.

**Table 1.** Biochemical compositions of *Desmodesmus* sp. in the form of percentages of the total dry biomass (DCW) at three different $CO_2$ concentrations for 12 days of cultivation time.

| Biochemical Composition | Inlet Concentration of $CO_2$ (*v/v*) | | |
|---|---|---|---|
| | 0.03% | 5% | 10% |
| Total carbohydrates (% DCW) | $14.6 \pm 1.5$ | $17.2 \pm 2.0$ | $20.7 \pm 2.4$ |
| Proteins (% DCW) | $14.4 \pm 1.2$ | $25 \pm 1.1$ | $32.3 \pm 2.5$ |
| Lipids (% DCW) | $15.5 \pm 0.5$ | $40 \pm 2.0$ | $42 \pm 1.0$ |
| Chlorophyll a, b (mg $L^{-1}$) | 0.12<br>0.15 | 0.13<br>0.17 | 0.14<br>0.19 |

The maximum protein content of $32.3 \pm 2.5$% DCW was obtained when algal cells were cultivated with 10% $CO_2$. The protein content was obtained as $14.4 \pm 1.2$% and $25 \pm 1.1$% when algal cells were grown at 0.03% and 5% $CO_2$, respectively (Table 1). The higher cell concentration with an increase in $CO_2$ concentration significantly increases the efficiency of the photosynthesis period. It leads to the formation of more and more amounts of protein.

The lipid content in algal biomass was observed to increase $CO_2$ concentration (Table 1). The maximum amount of lipid, about $42 \pm 1.0$% DCW, was accumulated at 10% $CO_2$. The lipid content was obtained as $15.5 \pm 0.5$% and $40 \pm 2.0$% for inlet $CO_2$ concentration of 0.03% and 5%, respectively. The scarcity of nitrogen and phosphorus due to their continuous utilization in microalgae cultivation may be the reason for higher lipid content [64].

The concentration of Chl a and Chl b were estimated using Equations (5) and (6), respectively, as is shown in Table 1 at different inlet $CO_2$ concentration of 0.03%, 5%, and 10% $CO_2$ (*v/v*). Chl a and Chl b concentration varies from $0.12 \pm 0.001$ to $0.14 \pm 0.004$ mg $L^{-1}$ and $0.15 \pm 0.002$ to $0.19 \pm 0.005$ mg $L^{-1}$ with the increase in inlet $CO_2$ concentration from 0.03% to 10%, respectively. The maximum amount of chlorophyll a and chlorophyll b was $0.14 \pm 0.004$ mg $L^{-1}$ and $0.19 \pm 0.005$ mg $L^{-1}$, respectively, when algae were treated with a 10% $CO_2$ concentration. It is observed that Chl a content is less than Chl b for all inlet $CO_2$ concentrations. It may be because the chlorophyll content in the microalgae varies in response to physical and chemical factors such as light intensity, agitation, temperature,

and nutrient availability [65,66]. In the present study, the microalgae species, *Desmodesmus*, is a genus of green algae in the family of *Scenedesmaceae*. In the green algae, chlorophyll b absorbs energy from wavelengths of green light at 640 nm, which may be a possible reason for a higher content of chlorophyll b in *Desmodesmus* sp. The concentration of chlorophyll a indicates the quantity and capacity of photosynthesis activity of microalgae. It can also be used to assess the physiological state of microalgae.

## 4. Performance Comparison of Loop Bioreactor

Table 2 shows the comparison of the performance of the custom-designed loop bioreactor with other reactors at different scales (bench, pilot, and large) in terms of the parameters such as biomass productivity ($g\,L^{-1}\,d^{-1}$), $CO_2$ fixation rate, and biochemical compositions (carbohydrate, protein, and lipid content) for $CO_2$ fixation via algal species. The maximum biomass produced ($1.903 \pm 0.038\,g\,L^{-1}$), biomass productivity ($0.19 \pm 0.004\,g\,L^{-1}\,d^{-1}$), and $CO_2$ fixation rate ($0.33 \pm 0.004\,g\,L^{-1}\,d^{-1}$) at 10% $CO_2$ concentration are higher or nearly the same as compared to the values reported for previous studies (Table 2). The carbohydrate ($20.7 \pm 2.4\%$) and protein ($32.3 \pm 2.5\%$) content obtained in the present study at 10% $CO_2$ concentration are comparable with the values reported for previous studies. However, the lipid content ($42 \pm 1.0\%$) is maximum compared to the studies reported in the literature. Most of the earlier studies were limited to the working volume of less than 1 L, except for a few studies. Compared to the large-scale bioreactors reported in the literature, the custom-designed loop bioreactor has shown better performance except for one study [35,67–69]. The scaled-up loop bioreactor has established comparable parameters that indicated the possibility of further scale-up of the process for the large-scale fixation of $CO_2$ and simultaneous algal biomass production, leading to by-product formation. This study may be a viable solution for the source reduction of $CO_2$ generated from waste management systems. The higher biomass productivity and carbohydrate content may lead to the value addition of the process in biofuels as by-product formation.

**Table 2.** Performance comparison of loop bioreactor in terms of various parameters with reported studies.

| Species | Cultivation Time (Day) | Mode/ (Volume, L/Working Volume, L) | $CO_2$ conc. (% *v/v*) | Max. Biomass Produced ($X_{Max}$) (g $L^{-1}$) | Biomass Productivity (P) (g $L^{-1}$ $d^{-1}$) | $CO_2$ Fixation Rate ($R_{CO2}$) (g $L^{-1}$ $d^{-1}$) | Carbohydrate (% DCW) | Protein (% DCW) | Lipids (% DCW) | References |
|---|---|---|---|---|---|---|---|---|---|---|
| *Chlorella* sp. | 8 | Column Photobioreactors, (0.8) | 2 | 1.21 | 0.15 | 0.28 | - | - | - | Chiu et al., 2008 [67] |
| *Chlorella vulgaris* | 15 | Bio Flow fermenter, (11/8) | 10 | 1.94 | 0.13 | 0.25 | 16.74 | 40.95 | 9.95 | Sydney et al., 2010 [70] |
| *Scenedesmus obliquus* | 6 | Erlenmeyer flask, (0.650) | 10 | 1.84 | 0.15 | 0.29 | - | - | 22 | Tang et al., 2011 [45] |
| *Chlorella sorokiniana* | 8 | Airlift photobioreactor, (1.4) | 4 | 1.1 | 0.15 | - | - | - | 20.93 | Kumar et al., 2014 [71] |
| *Scenedesmus* sp. | 7 | Airlift photobioreactor, (0.5) | 2.5 | 1.3 | 0.19 | 0.35 | 10.4 | - | 35.6 | Nayak et al., 2016 [72] |
| *Scenedesmus* sp. | 7 | Bubble-column photobioreactor, (0.5) | 2.5 | 1.37 | 0.196 | 0.37 | - | - | 33.3 | Nayak et al., 2016 [72] |
| *Acutodesmus* sp. | 5 | Erlenmeyer flasks, (0.5/0.2) | 20 | 1.65 | - | - | 34.52 | 38.78 | 11.67 | Yadav et al., 2015 [55] |
| *A. quadricellulare* | 6 | Laboratory scale photobioreactor, (0.8/0.680) | 5 | 1.29 | - | - | 33.4 | 30.3 | 44 | Varshney et al., 2018 [53] |
| *Desmodesmus* sp. MCC34 | 18 | Raceway pond, (1000) | - | 1.9 | - | - | - | - | 0.103 | Nagappan et al., 2016 [39] |
| *Porphyridium cruentum* | - | Airlift tubular, (200) | - | 3.0 | 1.50 | - | - | - | - | Yen et al., 2015 [71] |
| *Chlorella sorokiniana* | - | Inclined tubular, (6.0) | 5 | 1.50 | 1.47 | - | - | - | - | Ugwu et al., 2002 [69] |
| *Arthrospira platensis* | - | Undular row tubular, (11) | - | - | 2.70 | - | - | - | - | Carlozzi P., 2003 [73] |
| *Phaeodactylum tricornutum* | 9 | Outdoor helical tubular, (75) | - | 2.95 | 1.40 | - | - | - | - | Hall et al., 2003 [74] |
| *Haematococcus pluvialis* | 16 | Bubble-column, (55) | - | 1.4 | 0.06 | - | - | - | - | Lopez et al., 2006 [75] |
| *Chlorella pyrenoidosa* | 1.25 | Tubular batch reactors, (0.660) | 10 | | 0.11 | 0.096 | - | - | - | Kargupta et al., 2015 [76] |
| *Chlorella PY-ZU1* | 4.5 | Cylindrical PBR (6) | 15 | - | 0.47 | 0.87 | - | - | - | Ye at al., 2018 [77] |
| *Desmodesmus* sp. | 12 | Loop photobioreactor, (34/26) | 0.03 | 0.96 ± 0.04 | 0.018 ± 0.002 | 0.013 ± 0.001 | 14.6 ± 1.5 | 14.4 ± 1.2 | 15.5 ± 0.5 | Present study |
| *Desmodesmus* sp. | 12 | Loop photobioreactor, (34/26) | 5 | 1.219 ± 0.04 | 0.084 ± 0.003 | 0.155 ± 0.003 | 17.2 ± 2.0 | 25 ± 1.1 | 40 ± 2.0 | Present study |
| *Desmodesmus* sp. | 12 | Loop photobioreactor, (34/26) | 10 | 1.903 ± 0.04 | 0.185 ± 0.004 | 0.333 ± 0.004 | 20.7 ± 2.4 | 32.3 ± 2.5 | 42 ± 1.0 | Present study |



## 5. Conclusions

Biofixation of $CO_2$ (g) at three different concentrations (0.03%, 5%, and 10% *v/v*) by *Desmodesmus* sp. was successfully demonstrated in the custom-designed loop photobioreactor. The maximum values of specific growth rate, biomass productivity, and $CO_2$ fixation rate were obtained as $1.903 \pm 0.04$ g L$^{-1}$, $0.19 \pm 0.004$ g L$^{-1}$ d$^{-1}$, and $0.333 \pm 0.004$ g L$^{-1}$ d$^{-1}$, respectively, at 10% $CO_2$ concentration. The higher values of carbohydrate ($20.7 \pm 2.4\%$), protein ($32.3 \pm 2.5\%$), and lipid ($42 \pm 1.0\%$) content at 10% $CO_2$ concentration confirmed the suitability of *Desmodesmus* sp. for the fixation of higher $CO_2$ concentrations. The concentration of Chl a indicated the possibility of more significant photosynthesis activity of *Desmodesmus* sp. It can be concluded from the comparison of the present study with the studies reported in the literature that the use of a scaled-up loop bioreactor could possibly be utilized for large-scale fixation of $CO_2$ emitted from waste management sources and reduces the problem of greenhouse gas emission. The more excellent biochemical constituents in algal biomass can also be utilized as potential feedstocks for biofuel applications. Thus, the present study leads to a waste-to-wealth process as a sustainable and eco-friendly strategy for biofuel component production with $CO_2$ sequestration.

**Author Contributions:** The present research work is interdisciplinary research work. Please acknowledge: Conceptualization, S.R. and S.G.; methodology, S.R., S.G. and S.K.V.; validation, S.G.; formal analysis, S.R., S.G. and A.A.; investigation, A.A., K.T. and A.K.; data curation, A.A., K.T. and S.G.; writing—original draft preparation, A.A. and S.R.; writing—review and editing, S.R. and S.G.; visualization, S.R. and S.K.V.; supervision, S.R.; project administration, S.G. and S.K.V.; funding acquisition, S.R. and S.G. All authors have read and agreed to the published version of the manuscript.

**Funding:** The corresponding author would like to thank Science and Engineering Research Board—Core Research Grant, SERB—CRG, India for granting the project in the related area (CRG/2018/002943).

**Institutional Review Board Statement:** Not Applicable.

**Informed Consent Statement:** Not Applicable.

**Data Availability Statement:** Data is available on request from the corresponding author.

**Acknowledgments:** The authors are thankful to Birla Institute of Technology and Science (BITS), Pilani campus, India for providing the facilities to carry out the detailed research work.

**Conflicts of Interest:** The authors declare no conflict of interest. The funders had no role in the design of the study; in the collection, analyses, or interpretation of data; in the writing of the manuscript, or in the decision to publish the results.

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
