# Peer review of "Bio-Mitigation of Carbon Dioxide Using Desmodesmus sp. in the Custom-Designed Pilot-Scale Loop Photobioreactor"

_sustainability, doi:10.3390/su13179882_

Round 1

Reviewer 1 Report

The manuscript “Bio-mitigation of carbon dioxide using Desmodesmus sp. in the indigenous designed pilot-scale loop photo bioreactor” focuses on CO2 mitigation by microalgae utilization.

The manuscript shows several inaccuracies (more or less important). The experimental design is not original and attractive. All data are presented without statistic and in some cases the figures cited in the text do not match the description. The work should be thoroughly revised and reorganized by authors. On this basis, I guess that it is not adequate for the publication in this journal.

Author Response

Reviewer – 1:

The manuscript “Bio-mitigation of carbon dioxide using Desmodesmus sp. in the indigenous designed pilot-scale loop photo bioreactor” focuses on CO2 mitigation by microalgae utilization.

The manuscript shows several inaccuracies (more or less important). The experimental design is not original and attractive. All data are presented without statistic and in some cases the figures cited in the text do not match the description. The work should be thoroughly revised and reorganized by authors. On this basis, I guess that it is not adequate for the publication in this journal.

Response:

The manuscript is thoroughly checked and revised as per the suggestions given by all reviewers. 

Reviewer 2 Report

Review of manuscript 1273433 submitted to MDPI Sustainability

The manuscript describes a custom photobioreactor (loop photobioreactor) designed by the authors and presents the results of a 12-day experiment to monitor the growth and CO2 fixation by the green alga Desmodesmus sp. at 3 CO2 concentrations (0.03%, 5% and 10%). The authors measured biomass production (dry weight), growth (optical density) and calculated CO2 fixation based on these measurements. They also measured the chlorophyll a, carbohydrate, protein and lipid content of the algal biomass at the end of the experiment. Highest biomass production and CO2 fixation, as well as maximum carbohydrate, proteins and lipid content,  was achieved at 10% CO2 concentration.

Overall comments

Although an important part of the study is the presentation of a new design of a photobioreactor, the authors fail to convey the novelty of it compared to other scaled-up photobioreactors. This should be more clearly presented in the introduction, which should explicitly mention, which types of photobioreactors are currently used for large-scale biomass production, what are the disadvantages and what are the features of the new loop photobioreactor that overcome the potential problems of previous designs. Also, it seems that not many studies have been performed using Desmodesmus. If this is a novelty of the study then it should be pointed out.

As a second overall comment, I think the discussion is not fulfilling its purpose. It is not so much about the limited length as the lack of proper comparative data from previous studies of similar design. The authors present in Table 1, literature data however all the other studies were conducted at bench scale. There are studies in the literature conducted at larger scale that this study could be compared to. Until this is presented clearly, we cannot conclude on the advantages of this system. Studies conducted with Desmodesmus should be included too whether bench- or pilot-scale.  

Detailed comments

  1. It is not clear to me why the authors analyse in particular the municipal solid waste and agricultural waste in the introduction. The described system could be used in any situation, any kind of CO2 emission and the intro should be kept general. This also appears in the abstract for no apparent reason.
  2. GHG appears in line 31 for the first time but it is explained in full further down in line 43.
  3. Please check ref. 13 for lines 48-50. It does not seem to be the original reference that should be used for this statement.
  4. Line 52. 6.4oC is an extreme number that I have never seen before. Checking the ref you are providing, I’ve only found a figure (TS.15) that is showing such an increase of one model only for 2300. It would be best to provide a range of temperatures given by all models and preferably up to 2100.
  5. Lines 54-56: I suggest to avoid comparisons between CCS and CCU in 3 lines. Best to merely mention that this study is about CCU. Same applies for next paragraph. It would be best to use this space and provide a more comprehensive review of current CCU approaches with microalgae. I would also recommend to remove Figure 1 and mention the take-home message in the text.
  6. Line 87: The word “indigenous” is not correct here. It should be replaced throughout with a different one e.g. custom design. Also, since the design of a new loop photobioreactor is a key result of this study, this should not be presented in the Introduction.
  7. Line 132: probably 1250 ml instead of L
  8. Line 157: please write w.r.t in full
  9. Line 158: Please write in italics the name of algal species throughout the manuscript.
  10. Line 173: Equation 2, should cell concentration be used here instead of biomass and how was cell concentration calculated from OD?
  11. Equation 3: “Fixation” instead of “fixatation”. Please explain how this equation (1.88) was calculated. How did you use the molecular weight of algae?
  12. Please check the headings and subheadings in Materials and Methods. It should be 2 instead of 1 and all subheadings are currently 1.1.
  13. Line 196: after every “sampling point” instead of “study”.
  14. Line 228: “with slight modifications” instead of “insignificant alterations”.
  15. Lines 246-248: This sentence is somehow problematic. For one thing, the lag phase is “short”, if you mention “shorter” then it should be compared to something else. Is it shorter that lag phases of other microalgae? In any case, this is the Results section and only the results of this study should be presented here. Any conclusions/comparisons should be in the Discussion.
  16. Line 256-257: Same here, should be in Discussion.
  17. Lines 266-267: OD does not have a unit (nm refers to the wavelength the measurement was made).
  18. Lines 271-274: again this should not be in the Results.
  19. Figure 5. Please use dark or Light in the legend instead of morning-Evening
  20. Lines 279-281: This was mistakenly left here I suppose.
  21. Lines 289-290: As above, comparisons to other studies should be done in the Discussion.
  22. 6. Please either use different colors for the bars or completely different patterns.
  23. Line 306: spelling error in algae name
  24. Lines 316-317: as above, should be in Discussion
  25. Figures 7-10. I strongly encourage the authors to present all these data in a single table instead of 4 different figures.
  26. Discussion/Conclusions: as I mentioned in my overall comments, It does not become clear what the advantages of the loop bioreactor are, compared to other tubular bioreactors, and if higher biomass production can be achieved because of this design. The growth parameters measured must be compared to those of similar-scale bioreactors. In reference 20, such comparisons among large-scale photobioreactors are presented and these references should be included.

Author Response

Reviewer – 2:

The manuscript describes a custom photobioreactor (loop photobioreactor) designed by the authors and presents the results of a 12-day experiment to monitor the growth and CO2 fixation by the green alga Desmodesmus sp. at 3 CO2 concentrations (0.03%, 5% and 10%). The authors measured biomass production (dry weight), growth (optical density) and calculated CO2 fixation based on these measurements. They also measured the chlorophyll a, carbohydrate, protein and lipid content of the algal biomass at the end of the experiment. Highest biomass production and CO2 fixation, as well as maximum carbohydrate, proteins and lipid content, was achieved at 10% CO2 concentration.

Overall comments

Although an important part of the study is the presentation of a new design of a photobioreactor, the authors fail to convey the novelty of it compared to other scaled-up photobioreactors. This should be more clearly presented in the introduction, which should explicitly mention, which types of photobioreactors are currently used for large-scale biomass production, what are the disadvantages and what are the features of the new loop photobioreactor that overcome the potential problems of previous designs. Also, it seems that not many studies have been performed using Desmodesmus. If this is a novelty of the study then it should be pointed out.

As a second overall comment, I think the discussion is not fulfilling its purpose. It is not so much about the limited length as the lack of proper comparative data from previous studies of similar design. The authors present in Table 1, literature data however all the other studies were conducted at bench scale. There are studies in the literature conducted at larger scale that this study could be compared to. Until this is presented clearly, we cannot conclude on the advantages of this system. Studies conducted with Desmodesmus should be included too whether bench- or pilot-scale.  

Response:

Thanks for your suggestions. As per the suggestion, novelty of the designed bioreactor was included in comparison to other scaled-up bioreactor in the introduction section. The performance of custom designed reactor was compared with bench and pilot scale reactors. The earlier studies with Desmodesmus were included in comparison section.  All changes have been highlighted by yellow color in the revised manuscript.

Detailed comments

  1. It is not clear to me why the authors analyse in particular the municipal solid waste and agricultural waste in the introduction. The described system could be used in any situation, any kind of CO2 emission and the intro should be kept general. This also appears in the abstract for no apparent reason.
  2. GHG appears in line 31 for the first time but it is explained in full further down in line 43.
  3. Please check ref. 13 for lines 48-50. It does not seem to be the original reference that should be used for this statement.
  4. Line 52. 6.4oC is an extreme number that I have never seen before. Checking the ref you are providing, I’ve only found a figure (TS.15) that is showing such an increase of one model only for 2300. It would be best to provide a range of temperatures given by all models and preferably up to 2100.
  5. Lines 54-56: I suggest to avoid comparisons between CCS and CCU in 3 lines. Best to merely mention that this study is about CCU. Same applies for next paragraph. It would be best to use this space and provide a more comprehensive review of current CCU approaches with microalgae. I would also recommend to remove Figure 1 and mention the take-home message in the text.
  6. Line 87: The word “indigenous” is not correct here. It should be replaced throughout with a different one e.g. custom design. Also, since the design of a new loop photobioreactor is a key result of this study, this should not be presented in the Introduction.
  7. Line 132: probably 1250 ml instead of L
  8. Line 157: please write w.r.t in full
  9. Line 158: Please write in italics the name of algal species throughout the manuscript.
  10. Line 173: Equation 2, should cell concentration be used here instead of biomass and how was cell concentration calculated from OD?
  11. Equation 3: “Fixation” instead of “fixatation”. Please explain how this equation (1.88) was calculated. How did you use the molecular weight of algae?
  12. Please check the headings and subheadings in Materials and Methods. It should be 2 instead of 1 and all subheadings are currently 1.1.
  13. Line 196: after every “sampling point” instead of “study”.
  14. Line 228: “with slight modifications” instead of “insignificant alterations”.
  15. Lines 246-248: This sentence is somehow problematic. For one thing, the lag phase is “short”, if you mention “shorter” then it should be compared to something else. Is it shorter that lag phases of other microalgae? In any case, this is the Results section and only the results of this study should be presented here. Any conclusions/comparisons should be in the Discussion.
  16. Line 256-257: Same here, should be in Discussion.
  17. Lines 266-267: OD does not have a unit (nm refers to the wavelength the measurement was made).
  18. Lines 271-274: again this should not be in the Results.
  19. Figure 5. Please use dark or Light in the legend instead of morning-Evening
  20. Lines 279-281: This was mistakenly left here I suppose.
  21. Lines 289-290: As above, comparisons to other studies should be done in the Discussion.
  22. Please either use different colors for the bars or completely different patterns.
  23. Line 306: spelling error in algae name
  24. Lines 316-317: as above, should be in Discussion
  25. Figures 7-10. I strongly encourage the authors to present all these data in a single table instead of 4 different figures.
  26. Discussion/Conclusions: as I mentioned in my overall comments, It does not become clear what the advantages of the loop bioreactor are, compared to other tubular bioreactors, and if higher biomass production can be achieved because of this design. The growth parameters measured must be compared to those of similar-scale bioreactors. In reference 20, such comparisons among large-scale photobioreactors are presented and these references should be included.

Responses:

Thanks for providing the constructive suggestion and observations. All changes have been incorporated and highlighted with yellow color in the revised manuscript. Few major responses are given below:

  • As the manuscript is submitted to the special issue, the work carried out has been connected with municipal solid waste and agricultural waste management.
  • As per the suggestions, the “Results” section has been renamed as “Results and Discussion” and “Comparison” section is renamed as “Performance comparison of loop bioreactor” to provide more clarity for readers.
  • Heading numbers have been corrected.
  • Results given on the Figures 7 – 10 has been combined and represented in Table – 1 of the revised manuscript.
  • As per the suggestion, few studies included in Table – 2 to compare the growth parameters of present study with similar-scale bioreactors reported in the literature.

Reviewer 3 Report

Highlight changes in yellow in a next revision, please. No track changes.

Consider comments in the entire text.

Please define all chemical formulas, because they are used later:

"carbon dioxide, methane, nitrous oxide"

Why links on the pdf?

"Thus, municipal solid waste and"

Consider checking subscripts and removing outputs (added in the text):

"Figure 1. A conceptual micro algal system for combined biofuels production, CO2 bio-mitigation. Inputs: carbon source, 67 CO2, nitrogen and phosphorus sources, energy source, solar energy. Outputs: value-added bio products and biofuels."

Assure no references means complete originality and remove if not the case and nor adapted/modified:

"Figure 2. Schematic diagram of Loop photo bioreactor."

See the gray under the text

either spacing always or not, check (it should have... international unit system):

"(0.03 %)"

Any reference suggested in the methods section to contextualize the research strategy?

Check punctuation: “semi-continuous mode”

Revise headings numbering…

1.1. Experimental procedure

1.1. Measurement of biomass growth rate

Etc

Until the very end…

Known data must be preceded by citations immediately before presentation:

Check equations

Do not use the term “formula”

Check all italics

References outside the introduction should include a direct reference to authors name, so to be clear why is the references included:

“0–1.0 mg mL−1) [38].”

Etc…

HUGE text: “Figure 4. Biomass conc. (gL-1) at three different CO2 concentrations (0.03 %, 5 % and 10 % v/v) over 260 the incubation time of 12 days.”

As several other cases

Authors should try to “upgrade” the formats, such as:

Figure 6. Effect of three different CO2 concentrations ( 0.03 %, 5 % and 10 % v/v) on the different growth kinetics parameters 304 (specific growth rate (d-1), biomass productivity (gL-1d-1), CO2 fixation rate (gL-1d-1)) of microalgae.”

Outdated style…

And more…

Figures 7 to 10 (distorted…) should be grouped.

There is no need to repeat similar aspects in figures, group then and add separate subcaptions, as in other cases… To be relevant…

Comparison of what?!

Either merge results with discussion, our just discussion…

4. Comparison & discussion

And remove the “&” everywhere…

This is a tiny section, so it should be merged

See that if a discussion there is a scarcity of references here…

The comparison is made by the authors in the text, remove the term from the caption, instead focus in making the table as relevant as possible, addling authors names also before reference number, as usual…

Table 1. Comparison of maximum biomass concentration,”

Conclusions:

As in an abstract, it should contain brief contextualization (to defend the study) and methodology, main findings and practical implications…

Look at title and abstract too (same structure, but different content)

I saw none:

Supplementary Materials: The following are available online at www.mdpi.com/xxx/s1, Figure S1: 377 title, Table S1: title, Video S1: title.”

References, check the format…

“Ramanathan V. The Greenhouse Theory of Climate Change : H-dIT. Sci 240 1988;4850:293–300.”

I would like to see more references from 2021

The manuscript is well written. If focusing in specific comments above, it can be improved.

Highlight novelty and implications

Last, but not the least:

Similarity check shows some relevant similarity in almost every part of the text that should be addressed, including parts which should mostly rely on originality.

That is a problem, to me.

Author Response

Highlight changes in yellow in a next revision, please. No track changes.

 Consider comments in the entire text.

 Please define all chemical formulas, because they are used later:

"carbon dioxide, methane, nitrous oxide"

 Why links on the pdf?

"Thus, municipal solid waste and"

 Consider checking subscripts and removing outputs (added in the text):

"Figure 1. A conceptual micro algal system for combined biofuels production, CO2 bio-mitigation. Inputs: carbon source, 67 CO2, nitrogen and phosphorus sources, energy source, solar energy. Outputs: value-added bio products and biofuels."

 Assure no references means complete originality and remove if not the case and nor adapted/modified:

"Figure 2. Schematic diagram of Loop photo bioreactor."

 See the gray under the text either spacing always or not, check (it should have... international unit system): "(0.03 %)"

 Any reference suggested in the methods section to contextualize the research strategy?

 Check punctuation: “semi-continuous mode”

 Revise headings numbering…

1.1. Experimental procedure

1.1. Measurement of biomass growth rate

Etc Until the very end…

 Known data must be preceded by citations immediately before presentation:

Check equations

Do not use the term “formula”

Check all italics

 References outside the introduction should include a direct reference to authors name, so to be clear why is the references included:

“0–1.0 mg mL−1) [38].”

Etc…

HUGE text: “Figure 4. Biomass conc. (gL-1) at three different CO2 concentrations (0.03 %, 5 % and 10 % v/v) over 260 the incubation time of 12 days.”

As several other cases

 Authors should try to “upgrade” the formats, such as:

Figure 6. Effect of three different CO2 concentrations (0.03 %, 5 % and 10 % v/v) on the different growth kinetics parameters 304 (specific growth rate (d-1), biomass productivity (gL-1d-1), CO2 fixation rate (gL-1d-1)) of microalgae.”

Outdated style…

And more…

 Figures 7 to 10 (distorted…) should be grouped.

There is no need to repeat similar aspects in figures, group then and add separate subcaptions, as in other cases… To be relevant…

 Comparison of what?!

Either merge results with discussion, our just discussion…

4. Comparison & discussion

And remove the “&” everywhere…

 This is a tiny section, so it should be merged

 See that if a discussion there is a scarcity of references here…

 The comparison is made by the authors in the text, remove the term from the caption, instead focus in making the table as relevant as possible, addling authors names also before reference number, as usual…

Table 1. Comparison of maximum biomass concentration,”

 Conclusions:

As in an abstract, it should contain brief contextualization (to defend the study) and methodology, main findings and practical implications…

 Look at title and abstract too (same structure, but different content)

 I saw none:

Supplementary Materials: The following are available online at www.mdpi.com/xxx/s1, Figure S1: 377 title, Table S1: title, Video S1: title.”

 References, check the format…

“Ramanathan V. The Greenhouse Theory of Climate Change : H-dIT. Sci 240 1988;4850:293–300.”

 I would like to see more references from 2021

 The manuscript is well written. If focusing in specific comments above, it can be improved.

Highlight novelty and implications

 Last, but not the least:

Similarity check shows some relevant similarity in almost every part of the text that should be addressed, including parts which should mostly rely on originality.

That is a problem, to me.

Responses:

Thanks for providing in-depth and detailed suggestions. Authors have gone through all the suggestions made by reviewer and carefully incorporated in the revised manuscript. The changes made by author’s have been highlighted by yellow color in the revised manuscript. Few major responses are given below:

  • As per the suggestion of other reviewer, Figure 1 has been removed and its relevance has been included in the text.
  • Cited references have been thoroughly checked.
  • Heading numbers have been checked and corrected in the revised manuscript.
  • As per the suggestions, the “Results” section has been renamed as “Results and Discussion” and “Comparison” section is renamed as “Performance comparison of loop bioreactor” to provide more clarity for readers.
  • Authors have not uploaded any supplementary material.
  • Abstract and conclusions have been thoroughly checked and revised.
  • Novelty of the work has been highlighted in the introduction section.
  • Few more latest references have been added in the revised manuscript.

Round 2

Reviewer 1 Report

Dear authors,

I appreciate the improvements made on the manuscript, but they are not enough to make it publishable on this journal.

I give you some food for thought and suggestions to rearrange the work for another journal.

The manuscript is titled “Bio-mitigation of carbon dioxide using Desmodesmus sp. in the custom designed pilot-scale loop photo bioreactor”. In my opinion, the word bio-mitigation is inappropriate because: i) the experiment uses commercial CO2 gas; ii) no atmospheric CO2 bio-capture and storage system is envisaged. For this reasons I don't understand the link between your work and the CO2 bio-mitigation. In addition, the improvement of biomass concentration by varying CO2 concentration is an already widely explored topic.

Material and Methods: Some inaccuracies are present.

The OD measurement description is repeated in the section 2.1, 2.3, and 2.4.

The grown temperature is reported to be 26±1 (section 2.1) and 30-35 °C (section 2.3). Which is the right one?

The cultivation period results 10 days in the section 2.1 and 12 days in the sections 2.3 and result and discussion.

Result and discussion:

The fig 4 is not clear. The OD measurement seems to be obtained by the same culture subjected to L/D cycle. I assume that a noticeable difference between L and D time is unlikely. For further studies I suggest to compare OD with the cell number.

Tab 1: In general, in plants as well as in microalgae chlorophyll a is higher than b.  Reconsider the pigments measurement.

Table 2: This manuscript should be an original article and not a review.

The data discussion is poor.

Author Response

Dear Reviewer 1,

Please see the changes as suggested in the revised manuscript.

Thanks and Regards,

Reviewer 2 Report

The authors have made significant changes to the manuscript which is now much improved compared to the original submission. I only recommend a final check for typos and the English e.g. in line 56 "...is focusing on..." rather than "is emphasized".

Author Response

Dear Reviewer 2,

Please refer to the revised manuscript as we have worked on the typos and English language.

Thanks and Regards,

Reviewer 3 Report

Highlight changes in yellow in a next revision, please. No track changes.

Consider comments in the entire text.

I need detailed responses to EACH of my comments?

Not in bulk

Not done:

“Please define all chemical formulas, because they are used later:

"carbon dioxide, methane, nitrous oxide"

The grey is still present:

“"Figure 2. Schematic diagram of Loop photo bioreactor."

 See the gray under the text either spacing always or not, check (it should have... international unit system): "(0.03 %)"

“Distorted text“”

The “2” in CO2 needs to be smaller:

Figure 3. Effect of time on cell concentration (g L-1) at three different CO2 concentrations (0.03%, 279 5% and 10% v/v).”

And more…

Low quality:

Figure 4. Effect of light and dark cycle on Optical density (OD) with respect to time for three differ-296 ent CO2 concentration.”

And format differs a lot between figures: font, etc

Revise format:

Table 1: Biochemical compositions of Dsemodesmus sp. in the form of percentages of the total dry biomass (DCW) at three dif-358 ferent CO2 concentrations for 12 days of cultivation time.”

Why duplicate information? Not relevant…

4. Performance comparison of loop bioreactor

Table 2. Performance comparison of loop bioreactor in terms of various parameters with reported studies.”

References in Table 2 must be preceded by authors names, as usual

Table 2. Performance comparison of loop bioreactor in terms of various parameters with reported studies.”

Revise according to template:

Author Contributions:” using abbreviations identified in the affiliations

Added references were not highlighted.

Similarity % is still significant. It lowered 1%...

This is a problem.

Author Response

Dear Reviewer 3,

The suggestions are incorporated in the revised manuscript and the comment sheet is updated.

Thanks and Regards,

Round 3

Reviewer 1 Report

Dear authors,

I appreciated the changes made to the manuscript “Bio-mitigation of carbon dioxide using Desmodesmus sp. in the 2 custom-designed pilot-scale loop photo bioreactor”, for this reason I consider a minor revision before publication.

Some minor comments

Keywords: Check the keywords according to authors instruction.

Line 60-61 “The physical and chemical methods…………….CO2 storage.”. Add references.

Some suggestions

Francesco Nocito, Angela Dibenedetto, Atmospheric CO2 mitigation technologies: carbon capture utilization and storage, Current Opinion in Green and Sustainable Chemistry,Volume 21,2020,Pages 34-43

Shih-HsinHo, Chun-YenChen Duu-JongLee, Jo-ShuChang. Perspectives on microalgal CO2-emission mitigation systems — A review. Biotechnology Advances, Volume 29, Issue 2, 2011, Pages 189-198.

Line 62-64.  “Alternatively, biological methods………..photosynthetic processes” Add references.

Some suggestions.

Salbitani G, Barone CMA, Carfagna S (2019) Effect of bicarbonate on growth of the oleaginous microalga Botryococcus braunii. Int J Plant Biol 10:35-37.

Mountourakis, F.; Papazi, A.; Kotzabasis, K. The Microalga Chlorella vulgaris as a Natural Bioenergetic System for Effective CO2 Mitigation—New Perspectives against Global Warming. Symmetry 2021, 13, 997.

Vetrivel Anguselvi, Reginald Ebhin Masto, Ashis Mukherjee and Pradeep Kumar Singh (May 29th 2019). CO<sub>2</sub> Capture for Industries by Algae, Algae, Yee Keung Wong, IntechOpen, DOI: 10.5772/intechopen.81800. Available from: https://www.intechopen.com/chapters/65952

Author Response

Please refer the manuscript. The changes are incorporated. Thank You

Reviewer 3 Report

Highlight changes in yellow in a next revision, please. No track changes.

The quality of some figures is still terrible

Figure 5: why add % to every value if % is indicated in legend…?

I have been detailed in comments, the text is now batter, I hope the authors are able to learn and move ahead

Author Response

The suggestions are incorporated in the revised manuscript. Thank You
